# A streaming, certificate-based reduction for convex hull preservation in planar point sets

Oswaldo Cadenas  *

School of Computer Science and Digital Technologies, London South Bank University, London, United Kingdom

* cadenaso@lsbu.ac.uk

## Abstract

Convex hull computation on large planar point sets is commonly preceded by geometric filtering to reduce input size. Motivated by invariants used in incremental convex hull maintenance, we derive a streaming, certificate-based reduction that discards only points certified interior to the hull. We show that the reduction preserves the exact convex hull and operates in a single streaming pass with only local geometric operations. Experiments on synthetic and real-world datasets demonstrate substantial reduction, retaining between 5% and 11% of input points on average for synthetic distributions and below 1% on large real-world data under typical arrival order.

## Introduction

Convex hull computation is a fundamental operation in computational geometry, with applications in robotics, computer graphics, data visualisation, archaeology, linear programming and biology [1]. In practice, convex hull computation often appears as one stage in a larger pipeline, and the input frequently contains many interior points that never contribute to the hull. Therefore, it is common to apply geometric filtering prior to exact hull construction in order to reduce input size and improve overall performance.

In applications with dynamic and real-time data, such as collision detection, live object tracking, and sensor data processing, points may arrive continuously as a high-speed stream. For these systems, efficient online filtering methods that operate in a single pass and guarantee convex hull equivalence are particularly valuable. When reductions are substantial, filtering can yield end-to-end performance improvements of several times compared to direct hull construction. Thus, streaming algorithms are an important component of modern geometric processing systems [2].

A practical convex hull filter must satisfy three core requirements. First, it must preserve correctness by retaining all true hull vertices; discarding even a single

**Data availability statement:** The software, derived result files, and supporting dataset materials are available in the GitHub repository at https://github.com/oswcad/streaming-hull-filter and in Zenodo archives: Software, DOI: 10.5281/zenodo.19496559; Dataset, DOI: 10.5281/zenodo.19497433. The real-world experiments use publicly available New York City Taxi and Limousine Commission Yellow Taxi Trip Data (March–May 2025), available from https://www.nyc.gov/site/tlc/about/tlc-trip-record-data.page. The repository and archived dataset include the scripts and file information required to retrieve, preprocess, and reproduce the analyses.

**Funding:** The author(s) received no specific funding for this work.

**Competing interests:** The authors have declared that no competing interests exist.

extreme point invalidates the result. Second, it must achieve useful and quantifiable reduction in the candidate set size, otherwise the overhead of filtering outweighs its benefit. Third, it must operate efficiently under streaming constraints, ideally in linear time with manageable memory overhead. These requirements are reflected throughout the literature on convex hull preprocessing and streaming computation [2–4], yet many existing methods satisfy only a subset of them.

Classical geometric filtering techniques, such as the Akl–Toussaint heuristic [5], are typically presented as offline preprocessing steps applied to a complete point set. These methods rely on fixed directional extrema and static inner polygons, and are not naturally suited to unordered streaming input. Several authors have proposed alternative geometric pruning strategies to accelerate convex hull computation [3,6,7], but these approaches generally assume offline access to the data or rely on global geometric structure.

Streaming convex hull construction has also been studied in more formal streaming and external-memory models [2,8]. These methods aim to maintain or approximate the hull itself under strict memory and pass constraints. In contrast, the present work focuses on conservative input reduction: exact hull construction is deliberately deferred, and the streaming algorithm is required only to retain all points that may contribute to the hull.

In this paper, we derive a streaming, certificate-based reduction for convex hull preservation. The method is motivated by geometric invariants used in incremental convex hull maintenance, but does not attempt to maintain a correct hull online. Instead, we identify which aspects of incremental hull logic can be safely relaxed while preserving correctness. The resulting algorithm discards only points that are certified interior to the hull, retains all potential hull vertices, and operates in a single streaming pass with only local geometric operations.

Concretely, we first present a correctness-preserving reformulation of incremental convex hull maintenance for unordered streaming input, expressed in terms of upper and lower x-monotone chains [9]. This formulation is used solely as a conceptual scaffold and is not claimed as a novel convex hull algorithm. We then introduce a principled relaxation of this exact pipeline into a conservative geometric filter. The relaxation weakens online enforcement of convexity and ordering discipline while preserving geometric predicates, safety rules, and streaming constraints. Temporary geometric inconsistency is permitted, but a strict containment invariant is maintained, ensuring that the exact convex hull is preserved after final refinement.

The main contribution of this work is therefore a correctness-preserving streaming geometric filter derived as a relaxation of incremental convex hull maintenance. The filter does not rely on fixed directions, static inner polygons, global sorting, or multi-pass processing. Experimental evaluation on synthetic datasets demonstrates that the method achieves substantial reduction in practice, typically retaining between 5% and 11% of the input points under random arrival order. As expected for any one-pass streaming filter, adversarial arrival orders can suppress rejection and result in no reduction; however, correctness is preserved in all cases.

**Contributions**

The contributions of this work are as follows:

1. We propose a one-pass streaming convex hull filter with formal hull preservation guarantees.

2. We introduce a local *sandwich certificate* that enables interior-point rejection using only incremental convex envelope information.

3. We provide a proof that the filter preserves the exact convex hull for any input stream.

4. We characterise the empirical behaviour of the method across multiple geometric distributions and arrival orders, including adversarial monotone streams.

5. We provide a large-scale runtime comparison with the classical Akl–Toussaint heuristic under identical experimental conditions.

Unlike extremal batch heuristics, the proposed method is a conservative one-pass streaming filter: it does not maintain the hull online, but rejects only points certified interior by a local sandwich certificate, while preserving exact hull equivalence.

## 1. Preliminaries: Convex hulls and streaming setting

### 1.1 Problem formulation

Let $P = (p_1, p_2, \ldots, p_n)$ be a finite sequence of planar points $p_i \in \mathbb{R}^2$ presented as a stream in arbitrary order. For each $k \in \{0, 1, \ldots, n\}$, let

$$P_k = \{p_1, \ldots, p_k\}$$

denote the processed prefix, with $P_0 = \varnothing$. Let $CH(X)$ denote the convex hull of a finite point set $X \subseteq \mathbb{R}^2$.

The objective of the proposed reduction is to construct, in a single streaming pass, a retained subset $R \subseteq P$ such that

$$CH(R) = CH(P),$$

while ideally $|R| \ll |P|$ in typical cases.

The reduction must satisfy the following constraints:

1. **Streaming constraint:** Each point is processed exactly once in arrival order.

2. **Single-pass operation:** No global reordering or multi-pass processing is allowed.

3. **Correctness guarantee:** No point may be discarded unless it is certified to lie in the convex hull of previously retained points.

The reduction therefore acts as a conservative geometric filter whose sole correctness requirement is hull preservation. Exact hull computation is applied only once, after streaming, to the final retained set $R$.

Throughout this work, we consider a *streaming* input model in which points of $P$ arrive sequentially in an arbitrary, unordered fashion. The algorithm processes each point in a single pass and maintains only limited auxiliary state. No assumptions are made about the distribution, ordering, or geometric regularity of the input points.

Given an input set $P$ and a retained subset $R \subseteq P$, we say that $R$ *preserves the convex hull* of $P$ if

$$CH(R) = CH(P).$$

In this context, a reduction is considered *safe* if it discards only points that are guaranteed not to be vertices of the convex hull.

The objective of the proposed method is not to construct or maintain the convex hull explicitly during streaming, but rather to identify and discard points that can be certified as interior with respect to CH($P$), while retaining all points that may potentially contribute to the hull. As a consequence, the retained set $R$ may contain interior points, but must contain all hull vertices of $P$.

No claims regarding optimality, efficiency of hull construction, or asymptotic speedup of convex hull algorithms are made at this stage. The focus of this section is solely to establish notation, assumptions, and the correctness criterion used throughout the paper. We now briefly review geometric invariants commonly used in incremental convex hull maintenance, which provide the conceptual foundation for the reduction proposed in this work. These invariants are discussed solely to motivate the design of safe certificates and do not imply that an exact convex hull is maintained during streaming.

The underlying ideal objective is to minimise the retained set subject to exact hull preservation and one-pass streaming constraints; the proposed method is a safe conservative approximation to that constrained reduction problem.

## 1.2  Online hull invariants

Our filtering approach builds on established incremental convex hull methods that maintain two x-monotone chains during online point insertion [9–11]. Rather than developing a novel algorithm, we adapt these well-studied techniques to create geometric constraints suitable for real-time data filtering.

**1.2.1  Maintaining chain invariants.**  The core idea follows standard incremental hull construction: we maintain both an upper chain (left-turning) and a lower chain (right-turning), each sorted by x-coordinate. As points arrive in arbitrary order, each is inserted into both chains at the position determined by binary search on x-coordinate. This x-monotone property is central to the approach, as it allows local convexity testing without requiring global hull recomputation.

After insertion, we repair local convexity through a straightforward process. For the upper chain, we require that consecutive triples of points ($p_{i-1}$, $p_i$, $p_{i+1}$) satisfy orient($p_{i-1}, p_i, p_{i+1}$) > 0 (counterclockwise turn). The lower chain enforces the symmetric constraint with orient($p_{i-1}, p_i, p_{i+1}$) < 0 (clockwise turn). When this condition is violated at any position, we remove the middle vertex and repeat the test until convexity is restored.

Critically, the newly inserted point $p$ may itself become a "bad" middle vertex during this repair process. If removing $p$ restores convexity, we reject it from that chain. A point is accepted as a hull candidate only if it survives in at least one of the two chains. This rejection mechanism provides the geometric filter we exploit: points that would lie strictly interior to the current hull are automatically excluded without explicit inside-outside testing.

Importantly, this rejection is with respect to the locally maintained chains at the time of insertion, and does not require that the hull be globally correct at every step. The correctness of the final hull follows from the fact that any point removed as a bad middle vertex cannot be an extreme point of the convex hull.

In practice, exact online maintenance of these invariants under arbitrary arrival order is facilitated by imposing a limited ordering discipline. In our formulation, points are processed in small bounded chunks, within which a fixed local ordering is applied before insertion. This bounded reordering does not affect correctness; the same hull is obtained regardless of the order in which points within a chunk are processed; but it improves stability by reducing pathological insertion sequences and accelerating the emergence of reliable visibility certificates.

The combination of x-monotone insertion, local convexity repair, and bounded local ordering yields an exact streaming convex hull algorithm that matches classical monotone chain constructions while admitting a datapath-style implementation. However, this approach enforces convexity aggressively: every violation is immediately repaired, and points that fail to contribute to either chain are discarded as soon as a certificate is detected.

In the next section, we deliberately relax this enforcement. By weakening convexity repair and removing the requirement for local ordering, while preserving the underlying geometric predicates and safety rules, we derive a conservative

 

streaming reduction. The resulting filter no longer maintains a globally correct hull during streaming, but retains all potential hull vertices and defers exactness to a final refinement stage.

## 2. From hull invariants to safe reduction

The online hull construction described in the previous section provides a natural foundation for filtering, but its strict enforcement of convexity is unnecessarily strong for our purpose. We do not need to maintain an exact hull during processing; we only need to guarantee that true hull vertices are never discarded. This observation leads to a deliberate relaxation of the algorithm.

### 2.1 Relaxation strategy

We preserve the geometric structure of the incremental method but weaken its enforcement rules. Specifically:

1. **Retain**: Points are located in the upper and lower chains by x-coordinate using binary search, preserving x-monotonicity.

2. **Weaken**: Convexity repair is limited to a single local test. After insertion, only the immediate neighbors of the insertion point are examined, and at most one violating vertex on each side may be removed. Cascading repairs are deliberately suppressed.

3. **Remove**: The newly inserted point $p$ is never rejected as a bad middle vertex. Any point not explicitly discarded by the rejection test is retained in the candidate set.

Crucially, this relaxation removes enforcement, not logic. The same geometric predicates are evaluated as in exact hull maintenance, but violations are not exhaustively acted upon. By under-enforcing convexity in the envelopes while over-accepting points into the candidate set, we obtain a conservative filter.

### 2.2 Containment invariant

The relaxed filter maintains the fundamental guarantee that no extreme point of the input set $P$ is ever discarded. Hence every vertex of the convex hull CH($P$) is contained in the retained candidate set $R$.

This guarantee follows from two properties.

Conservative rejection.

A point is rejected only when a sandwich certificate can be constructed showing that it lies strictly between the current upper and lower envelopes. Given a point $p$ whose x-coordinate falls between existing envelope vertices, we locate its immediate neighbors in each chain and test whether $p$ lies below the upper envelope and above the lower envelope. Only points satisfying both conditions are discarded.

Because convexity enforcement is weakened, the envelopes may deviate from a true convex hull. This makes rejection less aggressive rather than more aggressive: any point that could potentially be a hull vertex will violate at least one of the sandwich conditions and will therefore be retained.

Selective insertion.

A point is inserted into an envelope only if it expands that envelope outward. Before insertion, we test whether $p$ would create a left turn (for the upper envelope) or a right turn (for the lower envelope) with its immediate neighbors. Points that lie inside the current envelope are not inserted, although they remain in the candidate set $R$.

Together, these mechanisms ensure that no true hull vertex is ever rejected, while still allowing substantial reduction in practice. The relaxed envelopes function as permissive geometric references rather than exact hull representations, trading precision in online structure for safety and simplicity in streaming reduction.

## 3. Hull preservation guarantee

For each $k \in \{0, 1, \ldots, n\}$, let $R_k \subseteq P_k$ denote the retained set after processing the first $k$ points, with $R_0 = \varnothing$. We write $x(q)$ and $y(q)$ for the coordinates of a point $q \in \mathbb{R}^2$.

Envelope segments and sandwich certificates.

After processing the prefix $P_{k-1}$, the algorithm maintains an upper envelope $U_{k-1}$ and a lower envelope $L_{k-1}$. Both are $x$-monotone polygonal chains whose vertices belong to $R_{k-1}$.

Let the incoming point be $p_k = (x_k, y_k)$. An *upper supporting segment* for $p_k$ is a segment $\overline{ab}$ joining two consecutive vertices $a,b$ of $U_{k-1}$ such that

$$x_k \in [\min\{x(a), x(b)\}, \max\{x(a), x(b)\}].$$

Similarly, a *lower supporting segment* for $p_k$ is a segment $\overline{cd}$ joining two consecutive vertices $c,d$ of $L_{k-1}$ such that

$$x_k \in [\min\{x(c), x(d)\}, \max\{x(c), x(d)\}].$$

Whenever such segments exist, let $u \in \overline{ab}$ and $\ell \in \overline{cd}$ denote the unique points on those segments with

$$x(u) = x(\ell) = x_k.$$

We say that $p_k$ satisfies the *sandwich certificate* if

$$y(\ell) < y_k < y(u).$$

Equivalently, $p_k$ lies strictly below the upper supporting segment and strictly above the lower supporting segment at the common abscissa $x_k$.

The algorithm discards $p_k$ only when both an upper supporting segment and a lower supporting segment exist and the sandwich certificate holds. Otherwise, $p_k$ is retained. If $x_k$ coincides with an envelope vertex shared by two adjacent segments, either incident segment may be used; the correctness argument is unchanged.

**Lemma 1 (Certificate implies convex-hull membership).** *If $p_k$ is discarded at step $k$, then*

$$p_k \in \mathrm{CH}(R_{k-1}).$$

*More precisely, if $\overline{ab}$ and $\overline{cd}$ are the supporting segments used in the sandwich certificate for $p_k$, then*

$$p_k \in \mathrm{CH}(\{a, b, c, d\}) \subseteq \mathrm{CH}(R_{k-1}).$$

*Proof.* Since $p_k$ is discarded, by definition there exist an upper supporting segment $\overline{ab}$ of $U_{k-1}$ and a lower supporting segment $\overline{cd}$ of $L_{k-1}$ whose horizontal projections contain $x_k$. Let $u \in \overline{ab}$ and $\ell \in \overline{cd}$ be the points with $x(u) = x(\ell) = x_k$. The sandwich certificate gives

$$y(\ell) < y_k < y(u).$$

Because $u$ and $\ell$ have the same $x$-coordinate as $p_k$, the point $p_k$ lies on the vertical segment joining $\ell$ and $u$. Hence there exists $\theta \in (0, 1)$ such that

$$p_k = (1 - \theta)\ell + \theta u.$$

Since $u \in \overline{ab}$, there exists $\lambda \in [0, 1]$ such that

$$u = (1 - \lambda)a + \lambda b.$$

Similarly, since $\ell \in \overline{cd}$, there exists $\mu \in [0, 1]$ such that

$$\ell = (1 - \mu)c + \mu d.$$

Substituting these expressions into the previous equation yields

$$p_k = (1 - \theta)\big((1 - \mu)c + \mu d\big) + \theta\big((1 - \lambda)a + \lambda b\big),$$

which is a convex combination of $a,b,c,d$. Therefore

$$p_k \in \mathrm{CH}(\{a, b, c, d\}) \subseteq \mathrm{CH}(R_{k-1}),$$

because $a, b, c, d \in R_{k-1}$. ☐

**Lemma 2 (Prefix hull invariant).** *For every $k \in \{0, 1, \ldots, n\}$,*

$$\mathrm{CH}(P_k) = \mathrm{CH}(R_k).$$

*Proof.* We argue by induction on $k$.

For $k=0$, both $P_0$ and $R_0$ are empty, so the claim is trivial.

Assume that

$$\mathrm{CH}(P_{k-1}) = \mathrm{CH}(R_{k-1})$$

holds for some $k \geq 1$. We show that it also holds for $k$.

**Case 1: $p_k$ is retained.** Then

$$R_k = R_{k-1} \cup \{p_k\}.$$

Hence, using the induction hypothesis,

$$\mathrm{CH}(P_k) = \mathrm{CH}(P_{k-1} \cup \{p_k\}) = \mathrm{CH}(R_{k-1} \cup \{p_k\}) = \mathrm{CH}(R_k).$$

**Case 2: $p_k$ is discarded.** Then

$$R_k = R_{k-1}.$$

By Lemma 1,

$$p_k \in \mathrm{CH}(R_{k-1}).$$

Using the induction hypothesis,

$$CH(P_k) = CH(P_{k-1} \cup \{p_k\}) = CH(P_{k-1}) = CH(R_{k-1}) = CH(R_k),$$

because adjoining a point already contained in the convex hull does not change that hull.

Thus the claim holds in both cases, completing the induction. □

**Theorem 3 (Hull Preservation).** *Let P be a finite planar point set processed by the proposed streaming reduction, and let R denote the final retained set. Then*

$$CH(P) = CH(R).$$

*Proof.* Apply Lemma 2 with $k=n$. □

## 4. Streaming reduction algorithm

Algorithm 1 summarises the proposed streaming reduction. The algorithm processes points in a single pass, maintaining relaxed upper and lower envelopes. Points are discarded only when a local sandwich certificate proves they lie strictly between the envelopes. All other points are retained in a candidate set for final refinement.

**Algorithm 1 Relaxed Streaming Convex Hull Filter**

```
Require: Stream of points p₁, p₂, ..., pₙ
Ensure: Candidate set R preserving CH(P)
 1: U ← ∅                                              ▷ (Upper envelope (x-monotone))
 2: L ← ∅                                              ▷ (Lower envelope (x-monotone))
 3: R ← ∅                                              ▷ (Retained candidate points)
 4: for each incoming point p do  ▷ (— Rejection test —)
 5:     if p has supporting segments in both envelopes and satisfies the sandwich certificate then
 6:         continue                                   ▷ (Discard p (certified interior))
 7:     end if
 8:     R ← R ∪ {p}                    ▷ (Retain point) ▷ (— Selective insertion into envelopes —)
 9:     if p expands the upper envelope U then
10:         Insert p into U by x-coordinate
11:         Perform at most one local convexity repair
12:     end if
13:     if p expands the lower envelope L then
14:         Insert p into L by x-coordinate
15:         Perform at most one local convexity repair
16:     end if
17: end for
18: return R
```

### 4.1 Complexity analysis

The proposed streaming reduction processes each input point exactly once and performs only bounded local work per point. For each incoming point, the algorithm evaluates a fixed number of orientation tests to determine rejection, and conditionally performs at most one insertion and one local convexity repair in each envelope.

Because cascading convexity repairs are explicitly suppressed, envelope maintenance involves no unbounded loops. All geometric predicates are evaluated in constant time, and no global sorting or batch operations are performed during streaming. Assuming the envelopes are stored in *x*-ordered dynamic search structures that support logarithmic

predecessor/successor queries and $O(1)$ local updates once the insertion position is located, the total streaming reduction runs in $O(n \log r)$ time, where $r$ is the final number of retained points.

As a result, the total time complexity of the reduction is $O(n \log r)$, where $r$ is the number of retained points. In the worst case, when no points are discarded and $r = n$, this yields a time complexity of $O(n \log n)$. The space complexity is linear in the size of the retained candidate set $R$, which in practice is substantially smaller than the input size, as demonstrated in Section 6. In the common practical scenario where the filter achieves substantial reduction ($r \ll n$, see Section 6), the observed running time is correspondingly close to linear.

Exact convex hull computation is applied only once to the reduced set after streaming completes.

This bound holds regardless of the degree of reduction achieved; even in adversarial cases where $r = n$ and no points are discarded, the filter processes each input point exactly once and performs only bounded local geometric operations.

## 5. Application context

Large-scale geometric processing pipelines frequently operate under streaming or memory-constrained conditions. Examples include LiDAR acquisition, real-time mapping, edge-device spatial sensing, and incremental surface reconstruction from point clouds [12]. Convex hull computation also plays a role in quantitative morphology and biological shape analysis. For example, convex hull volume has been used to study scaling relationships between body mass and skeletal geometry in primates and other vertebrates [1]. Such applications highlight the importance of efficient hull computation in scientific workflows involving large datasets.

In such settings, full convex hull construction may serve as a preprocessing step for downstream tasks, including:

- surface triangulation,

- outlier rejection,

- boundary detection,

- spatial partitioning and meshing,

- geometric summarisation.

Reducing the candidate set prior to hull construction can decrease memory usage and improve computational efficiency, particularly when data arrive incrementally or cannot be stored in full.

Unlike classical batch filters, the proposed method operates in streaming mode and therefore aligns naturally with pipelines where points are processed as they are acquired. Its conservative design ensures that downstream hull-dependent computations remain exact.

## 6. Experimental evaluation

This section presents an empirical evaluation of the relaxed streaming convex hull filter. The experiments serve two complementary purposes. First, we characterise the behavioural properties of the filter under controlled geometric and arrival-order variations. Second, we evaluate its large-scale reduction efficiency and runtime performance relative to classical pre-filtering heuristics.

The objective is to characterise practical performance under realistic streaming conditions rather than to optimise asymptotic bounds.

### 6.1 Reduction under random arrival

Fig 1 shows the mean reduction ratio under random arrival order for seven representative point distributions.

For most distributions, the filter achieves substantial reduction, retaining between 5% and 11% of the input points on average. Strong reduction is observed for uniform, Gaussian, banana-shaped, and clustered distributions. The annulus distribution exhibits moderate reduction, reflecting the higher proportion of points near the true hull boundary.

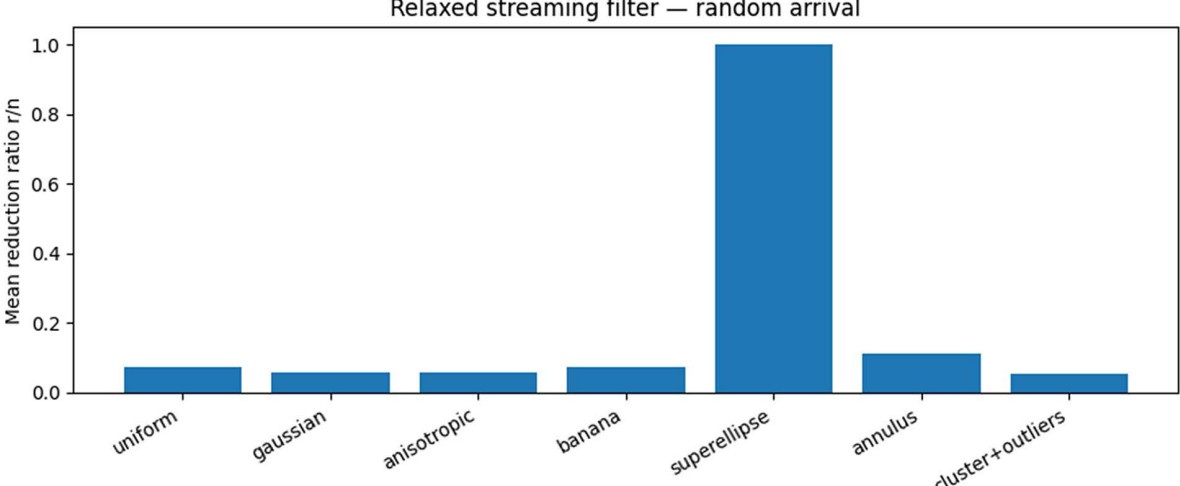

**Fig 1. Mean reduction ratio ($r/n$) achieved by the relaxed streaming convex hull filter under random arrival order, averaged over 20 independent runs.** Results are shown for seven geometric point distributions. Strong reduction is observed for most distributions, while no reduction occurs for the superellipse distribution due to its geometric ambiguity near the hull.

In contrast, the superellipse distribution exhibits no reduction. This behaviour is geometrically expected: sharp corners and extended flat regions generate many points that are locally consistent with being hull vertices. The conservative design of the streaming filter therefore retains these points rather than discarding them without sufficient local certification.

These results demonstrate that the method is effective under typical geometric conditions while remaining structurally conservative in ambiguous near-hull regimes.

### 6.2 Order sensitivity

Fig 2 illustrates the sensitivity of the relaxed streaming filter to arrival order for uniform and Gaussian distributions.

Under random arrival, strong reduction is consistently achieved. Under zigzag order, partial reduction remains possible, indicating robustness to moderately adversarial streaming patterns. However, for monotone arrival orders (sorted or reverse-sorted by $x$, and sorted by $y$), the filter performs no rejection and retains all points.

This behaviour reflects a structural limitation of single-pass streaming filters: when early points dominate envelope expansion, later points cannot be certified as interior using only local sandwich information. Importantly, correctness is preserved in all cases. The absence of rejection under monotone order therefore represents a conservative failure mode rather than an incorrect one.

These experiments clarify the operational envelope of the method and make explicit the trade-off between streaming constraints and rejection capability.

### 6.3 Large-scale comparison with the Akl–Toussaint heuristic

The preceding experiments characterise the structural behaviour of the relaxed streaming filter under controlled geometric and arrival-order variations. We now turn to a complementary question: how does the proposed method compare, in large-scale settings, to classical hull pre-filtering heuristics?

Among existing techniques, the Akl–Toussaint heuristic is a well-known and widely used pre-processing step for convex hull computation. It removes interior points by discarding those lying inside an extremal polygon constructed from a small

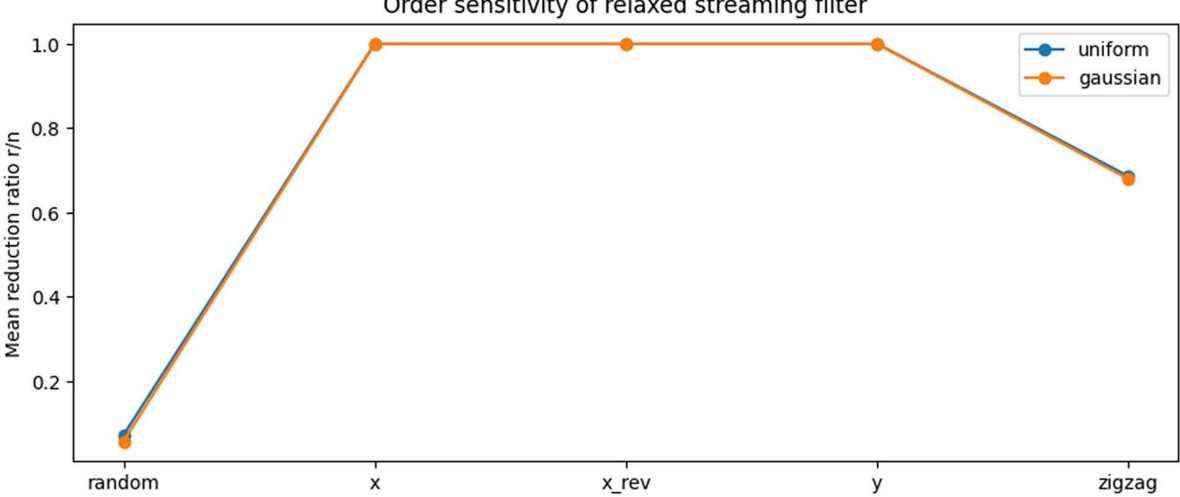

**Fig 2. Sensitivity of the relaxed streaming convex hull filter to arrival order.** Mean reduction ratio ($r/n$) is shown for uniform and Gaussian point distributions under five arrival orders. Substantial reduction is achieved under random arrival, partial reduction under zigzag order, and no reduction under monotone arrival orders. In all cases, exact hull preservation is maintained.

set of boundary directions. Unlike the proposed method, however, it is not designed for one-pass streaming operation and assumes full access to the input set.

We evaluate both methods under identical large-scale conditions. The comparison focuses on reduction ratio and total runtime relative to full hull computation, while recognising that the relaxed filter is explicitly designed for streaming environments rather than batch preprocessing.

This comparison quantifies the practical trade-offs between streaming capability and classical batch-based reduction strategies.

**6.3.1 Experimental setup.** All experiments were run in Python on an Apple laptop equipped with an M4 processor using double-precision arithmetic. Runtime measurements were obtained using `time.perf_counter()`, averaged over 20 independent trials. Each configuration includes a warm-up phase prior to timing to reduce interpreter and caching effects. For the NYC Taxi experiments, results are reported as mean ± standard deviation over 3 repetitions.

Convex hull computation uses a standard monotone chain implementation applied:

1. Directly to the full input set (baseline),

2. After filtering with the relaxed streaming method,

3. After filtering with the Akl–Toussaint heuristic.

We report total runtime (filter + hull) and compute speedup relative to full hull construction.

**6.3.2 Baseline: Akl–Toussaint heuristic.** As a classical convex hull pre-filter, we implement the Akl–Toussaint heuristic, which constructs a convex polygon from extremal $x$ and $y$ coordinates and discards points strictly inside this polygon. This method is widely cited in computational geometry as a linear-time hull reduction heuristic.

All implementations preserve exact hull correctness. No hull mismatches were observed in any experiment.

**6.3.3 Point distributions.** We consider four representative geometric distributions:

1. Uniform distribution in the unit square,

2. Isotropic Gaussian distribution,

3. Uniform distribution in the unit disk,

4. Superellipse distribution with sharp corners.

These distributions span easy, typical, and near-adversarial geometric regimes.
Experiments were conducted for

$$n \in \{10^4, 5 \times 10^4, 10^5, 5 \times 10^5, 10^6\}.$$

**6.3.4 Reduction efficiency.** Fig 3 illustrates the runtime speedup relative to full hull construction for the Gaussian distribution.

For random distributions (Gaussian, disk, square), the relaxed filter retains a rapidly decreasing fraction of points as $n$ increases, typically below 0.1% at $n = 10^6$. In contrast, the Akl–Toussaint heuristic retains a constant fraction for uniform square inputs and a small but non-negligible fraction for Gaussian inputs.

For the superellipse distribution, nearly all points lie close to the true hull boundary. Both filters retain most points in this case, demonstrating conservative behaviour under hull-dominated inputs.

**6.3.5 Runtime performance.** Table 1 summarises results at $n = 10^6$.

For Gaussian inputs, the relaxed filter achieves nearly 2× speedup over full hull construction at $n = 10^6$, with low variance across trials. The Akl–Toussaint heuristic provides moderate improvement for Gaussian data but little to no improvement for uniform square distributions.

In the superellipse case, where most points lie near the hull, both methods exhibit reduced speedup, reflecting the limited opportunity for safe interior certification. Importantly, correctness is preserved in all cases.

**6.3.6 Reduction behaviour.** Under random and chronological arrival, the streaming filter consistently reduced the candidate set size, retaining a small fraction of interior points while preserving all hull vertices. Reduction ratios remained

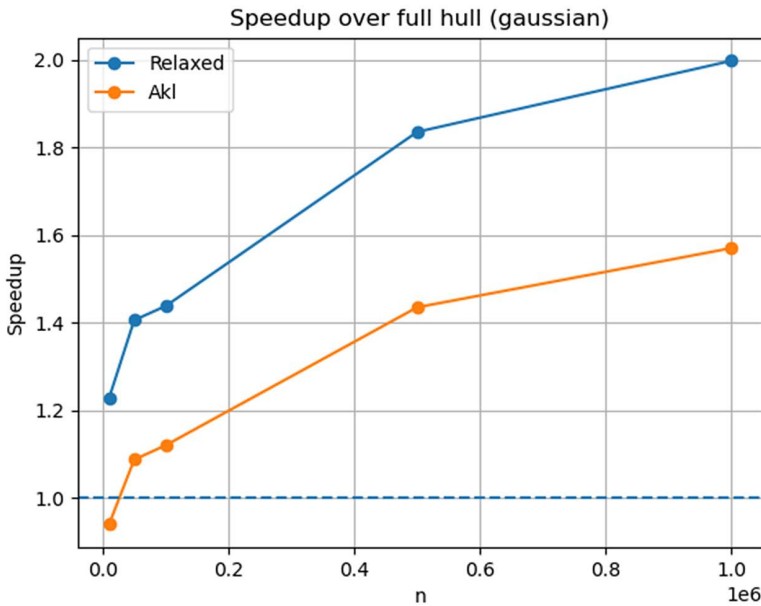

**Fig 3. Speedup achieved due to the reduction of points by relaxed streaming convex hull filter and Akl–Toussaint heuristic, averaged over 20 independent runs.** Results are shown for a Gaussian distribution over a sweep of $n$ from 10K to 1M points.

**Table 1. Mean retained ratio and speedup at $n = 10^6$ (20 trials).**

| Geometry | Relaxed $r/n$ | Akl $r/n$ | Relaxed Speedup | Akl Speedup |
|---|---|---|---|---|
| Disk | 0.0010 | 0.3635 | 1.55× | 1.06× |
| Gaussian | 0.00020 | 0.00498 | 2.00× | 1.57× |
| Square | 0.00033 | 0.5074 | 1.88× | 0.97× |
| Superellipse | 0.9988 | 1.0000 | 0.03× | 0.66× |

stable as $n$ increased, indicating scalability under clustered real-world spatial distributions. As expected, under longitude-sorted arrival order, little or no reduction occurred, confirming sensitivity to adversarial monotone streams.

**6.3.7 Observations.** Across all experiments, the following consistent patterns are observed:

- Exact hull preservation is maintained in all trials.

- The relaxed filter achieves strong reduction for random distributions, with retained fraction decreasing as $n$ increases.

- The Akl–Toussaint heuristic retains a constant fraction for uniform distributions.

- Runtime speedup scales with reduction efficiency.

- Performance degrades gracefully under hull-dominated inputs.

These results indicate that the proposed method functions as an effective conservative streaming filter that provides substantial practical acceleration under typical geometric conditions while maintaining transparency regarding its limitations.

## 6.4 Evaluation on real-world GPS data

To evaluate performance under realistic spatial distributions, we conducted experiments using publicly available New York City Taxi & Limousine Commission (TLC) trip data (March–May 2025) [13]. From each record, the pickup longitude and latitude were extracted to form two-dimensional point sets. Records with missing or invalid coordinates were discarded. The resulting datasets contain millions of spatial samples exhibiting strong clustering, directional structure, and sparse geographic outliers.

For scalability analysis, we considered subsamples of size

$$n \in \{50{,}000, \ 100{,}000, \ 250{,}000, \ 500{,}000\}.$$

Each subsample was evaluated under three arrival orders:

- Chronological order (natural streaming order),

- Random shuffle,

- Longitude-sorted order (adversarial monotone order).

For each configuration, we measured:

- Reduction ratio $r/n$ after streaming filtering,

- Total runtime of full hull computation,

- Runtime of Akl–Toussaint filtering followed by hull construction,

- Runtime of the proposed streaming filter followed by hull construction.

Exact hull preservation was verified by comparing the vertex sets of the hull computed on the full dataset and on the filtered candidate set. No discrepancies were observed in any experiment.

Table 2 summarises performance on the NYC Taxi dataset under random arrival order. The relaxed streaming filter achieves substantial reduction, with the retention ratio decreasing from $1.19 \times 10^{-3}$ at $n = 50{,}000$ to $1.25 \times 10^{-4}$ at $n = 500{,}000$. In contrast, the Akl–Toussaint heuristic retains approximately 5% of input points across all tested sizes. This difference translates into measurable runtime gains: at $n = 500{,}000$, the relaxed filter achieves a 1.63× speedup relative to computing the full hull directly, compared to 1.31× for Akl–Toussaint. The standard deviations are small for the relaxed filter across all tested sizes, indicating stable reduction and runtime behaviour under repeated random shuffling. Notably, the reduction strength of the relaxed filter improves with increasing $n$, whereas Akl–Toussaint exhibits near-constant retention.

Absolute wall-clock timings for the NYC Taxi experiments are provided in Supporting Information (S1 Table).

## Reproducibility

All code required to reproduce the experiments and figures in this paper is publicly available at https://github.com/oswcad/streaming-hull-filter. Permanent archived snapshots of the software and dataset are available at Zenodo: Software, DOI https://doi.org/10.5281/zenodo.19496559; Dataset, DOI https://doi.org/10.5281/zenodo.19497433. The repository includes scripts for synthetic data generation, experimental evaluation, figure production, and retrieval and preprocessing of the public NYC TLC datasets used in the real-world experiments.

## Discussion

The experimental results confirm that the proposed filter occupies a distinct position among convex hull preprocessing techniques. Unlike classical batch algorithms and extremal-based heuristics such as Akl–Toussaint, the method operates strictly in streaming mode while preserving exact hull correctness.

Across both synthetic distributions and the large-scale NYC taxi dataset, the filter consistently reduces the candidate set substantially under random and chronological arrival. On real-world taxi data comprising up to $5 \times 10^5$ points, the retained fraction drops below 0.02% under random order, compared to approximately 5% for Akl–Toussaint. This reduction translates directly into measurable runtime gains, with observed end-to-end speedups of approximately 1.6× relative to full hull construction, and consistent improvement over the Akl heuristic in typical (non-adversarial) orderings.

The sensitivity analysis clarifies the structural limits of conservative streaming filters. Monotone orderings reduce rejection capability because early envelope growth restricts interior certification; however, correctness is never compromised. Importantly, the taxi dataset experiments demonstrate that natural arrival patterns (chronological trip order) behave similarly to random order rather than adversarial constructions, indicating that the worst-case behaviour is unlikely in practical data streams.

From a theoretical standpoint, the local sandwich certificate distinguishes the method from extremal-based filtering: interior rejection is certified incrementally without global extrema or multiple passes. To the best of our knowledge, no

**Table 2. Performance comparison on the NYC Taxi dataset under random arrival order. Values are reported as mean ± standard deviation over 3 repetitions. Retention ratio denotes $r/n$. Speedup is measured relative to computing the full hull directly.**

| $n$ | Relaxed $r/n$ | Akl $r/n$ | Relaxed speedup | Akl speedup |
|---|---|---|---|---|
| 50,000 | $1.19 \times 10^{-3} \pm 5.8 \times 10^{-5}$ | $4.58 \times 10^{-2} \pm 2.1 \times 10^{-3}$ | $1.45 \pm 0.04$ | $1.00 \pm 0.01$ |
| 100,000 | $6.17 \times 10^{-4} \pm 6.0 \times 10^{-5}$ | $5.08 \times 10^{-2} \pm 3.6 \times 10^{-3}$ | $1.46 \pm 0.06$ | $1.11 \pm 0.13$ |
| 250,000 | $2.20 \times 10^{-4} \pm 2.9 \times 10^{-5}$ | $5.12 \times 10^{-2} \pm 5.8 \times 10^{-3}$ | $1.59 \pm 0.01$ | $1.28 \pm 0.17$ |
| 500,000 | $1.25 \times 10^{-4} \pm 1.3 \times 10^{-5}$ | $5.38 \times 10^{-2} \pm 2.6 \times 10^{-3}$ | $1.63 \pm 0.03$ | $1.31 \pm 0.16$ |

prior convex hull filter combines (i) strict one-pass streaming operation, (ii) formal proof of exact hull preservation, and (iii) empirical characterisation across both synthetic and real large-scale datasets including order sensitivity.

These results suggest that the filter is particularly suited to streaming geometric pipelines, memory-constrained processing, and incremental spatial analytics where batch preprocessing is infeasible. Future work may investigate adaptive certificates that mitigate order sensitivity while maintaining conservative guarantees, as well as extension to higher-dimensional streaming hull construction.

## 7. Related work

Convex hull computation is a classical problem in computational geometry, with well-known optimal algorithms achieving $O(n \log n)$ time in the algebraic decision tree model [10]. Graham scan, divide-and-conquer methods, and incremental algorithms form the traditional algorithmic foundation [14].

### 7.1 Output-sensitive and incremental methods

Output-sensitive algorithms, such as Chan's algorithm [15,16], achieve $O(n \log h)$ complexity, where $h$ is the number of hull vertices. Randomised incremental approaches and Clarkson–Shor style techniques further refine expected performance [17]. These methods improve theoretical efficiency but remain fundamentally batch algorithms requiring access to the full dataset.

### 7.2 Pre-filtering heuristics

Several heuristics aim to reduce input size prior to hull construction. The Akl–Toussaint heuristic [5] constructs a convex polygon from extremal coordinates and discards interior points. Variants extend this idea using additional directional extrema to improve reduction rates. Variants of pre-filtering have also been proposed for accelerating convex hull computation through data preconditioning, including integer-specific heuristics for 2D point sets [18,19] and GPU computing [4].

Grid-based and spatial binning techniques have also been used as practical pre-filters in large-scale geometric processing pipelines. However, such approaches typically require either multiple passes or global spatial indexing structures.

### 7.3 Streaming and one-pass geometry

Streaming geometric algorithms impose stronger constraints: each point must be processed once, in arrival order, without revisiting past data. While streaming variants exist for certain geometric problems (e.g., approximate hulls [20], bounding boxes, or sketch-based summaries), exact hull preservation under strict one-pass filtering constraints has received limited attention [2].

Most existing hull filters are designed for batch settings, where interior certification may rely on global knowledge of the dataset. In contrast, streaming settings prohibit reconstruction or reordering of the full point set during filtering.

### 7.4 Position of the present work

The present work differs from prior convex hull filters in two principal ways:

1. It operates strictly in one-pass streaming mode without requiring global spatial partitioning or multiple scans.

2. It provides a formal guarantee of exact hull preservation while performing interior rejection based solely on incremental envelope information.

To the best of our knowledge, no previously published method combines conservative hull preservation guarantees with single-pass streaming operation and empirical analysis of arrival-order sensitivity.

## Conclusion

We have presented a conservative one-pass streaming convex hull filter that reduces the candidate set prior to exact hull construction while guaranteeing exact hull preservation. Unlike classical batch algorithms and pre-filtering heuristics that require global access to the dataset, the proposed method operates strictly in streaming mode: each point is processed once in arrival order, and no reconstruction of the full point set is required during filtering.

The method is based on a local sandwich certificate that permits interior-point rejection using only incremental convex envelope information. We provided a formal proof that no true hull vertex is discarded, ensuring that $CH(R) = CH(P)$ for all inputs.

Within the broader convex hull literature, classical and output-sensitive algorithms achieve optimal asymptotic bounds in batch settings, and the Akl–Toussaint heuristic performs effective global pre-filtering. In contrast, this work addresses conservative filtering under strict streaming constraints. To the best of our knowledge, no previously published method combines one-pass operation, exact hull preservation, formal correctness guarantees, and empirical characterisation of arrival-order sensitivity in this setting.

Empirically, substantial reduction is observed under typical random arrival across synthetic and real-world datasets, including large-scale NYC taxi data, leading to measurable runtime improvements relative to both full hull construction and the Akl–Toussaint heuristic at increasing scale. As expected, adversarial monotone arrival suppresses rejection, reflecting intrinsic limitations of local streaming certificates rather than algorithmic failure. In all cases, exact hull correctness is maintained.

These results position the proposed filter as a practical preprocessing component for streaming and memory-constrained geometric pipelines, including point cloud processing and incremental spatial analytics. Future work may explore adaptive certificates that increase robustness to adversarial orderings while preserving conservative guarantees, as well as extensions to higher dimensions.

## Supporting information

**S1 Table. Absolute runtime (s) on the NYC Taxi dataset under random arrival order.** Values are reported as mean ± standard deviation over 3 repetitions. Timings were obtained from Python reference implementations on an Apple laptop with an M4 processor and are reported for transparency rather than as hardware-optimised benchmarks.
(PDF)

## Acknowledgments

The author used ChatGPT (OpenAI) as an assistive tool for language refinement, stylistic refinement, and clarification of presentation. The AI tool was not used to generate scientific results, proofs, algorithms, experimental data, or interpretations. All technical content, experimental design, and conclusions were developed and verified by the author, who reviewed and validated all AI-assisted text for accuracy and consistency.

## Author contributions

**Conceptualization:** Oswaldo Cadenas.

**Formal analysis:** Oswaldo Cadenas.

**Investigation:** Oswaldo Cadenas.

**Methodology:** Oswaldo Cadenas.

**Software:** Oswaldo Cadenas.

**Validation:** Oswaldo Cadenas.

**Visualization:** Oswaldo Cadenas.

**Writing – original draft:** Oswaldo Cadenas.

**Writing – review & editing:** Oswaldo Cadenas.

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
