## [Decision Letter · Decision Letter 0]

31 Mar 2026

PONE-D-26-02494A Streaming, Certificate-Based Reduction for Convex Hull Preservation in Planar Point SetsPLOS One

Dear Dr. Cadenas,

Thank you for submitting your manuscript to PLOS ONE. After careful consideration, we feel that it has merit but does not fully meet PLOS ONE’s publication criteria as it currently stands. Therefore, we invite you to submit a revised version of the manuscript that addresses the points raised during the review process.

**ACADEMIC EDITOR:** The authors are required to further elaborate on the problem description, algorithm design and other methodological details.

We look forward to receiving your revised manuscript.

Kind regards,

Jiapeng Liu, Ph.D.

Academic Editor

PLOS One

**Journal Requirements:**

2. Please note that your Data Availability Statement is currently missing the repository name. If your manuscript is accepted for publication, you will be asked to provide these details on a very short timeline. We therefore suggest that you provide this information now, though we will not hold up the peer review process if you are unable.

3. Please remove your figures from within your manuscript file, leaving only the individual TIFF/EPS image files, uploaded separately. These will be automatically included in the reviewers’ PDF.

Reviewers' comments:

Reviewer's Responses to Questions

**Comments to the Author**

1. Is the manuscript technically sound, and do the data support the conclusions?

Reviewer #1: Yes

Reviewer #2: Yes

Reviewer #3: No

Reviewer #4: No

2. Has the statistical analysis been performed appropriately and rigorously?

Reviewer #1: Yes

Reviewer #2: No

Reviewer #3: No

Reviewer #4: No

3. Have the authors made all data underlying the findings in their manuscript fully available?

Reviewer #1: Yes

Reviewer #2: Yes

Reviewer #3: Yes

Reviewer #4: Yes

4. Is the manuscript presented in an intelligible fashion and written in standard English?

Reviewer #1: Yes

Reviewer #2: Yes

Reviewer #3: No

Reviewer #4: Yes

5. Review Comments to the Author

Reviewer #1: The paper needs a fully formal correctness proof, comparisons with existing convex hull filters, tests on real-world and larger datasets, an analysis of sensitivity to input order, and concrete runtime measurements to show practical benefits.

Reviewer #2: The paper comes across as technically sound, but very narrowly focused on the particulars of the filter design to achieve the aim of the paper; it would improve the impact of this work if the context information was provided as well; e.g. how this connects with other processes involved in this type of surface reconstruction, what measurable benefits are being identified and how they can be applicable to a broader context of this topic. In addition, the statistical analysis is very sparse - it could be more informative and it should be supported by graphical representation, which is very modest at present. Also, the access to data was declared as entirely available, but it is not obvious to the reviewer how and where from the data can be accessed. As it stands. it leaves the reader wanting more information. Overall, with the appropriate amendments, I believe the paper could be publishable

Reviewer #3: After carefully evaluating the manuscript, I did not find enough novelty or originality to support publication. The study does not demonstrate a clear conceptual, methodological, or practical advancement over existing literature. Since the manuscript lacks a strong innovative contribution and its significance appears limited, I recommend rejection.

Reviewer #4: 1. 12 citations are clearly not enough. More references must be discussed, and please add a more detailed literature review. Author must explicitly show the place of this research in the general context of this field. More details should be furnished.

2. Problem formulation is quite vague. The mathematical aspects of this proposal must be presented in detail to enhance the scientific rigor of the paper. The mathematical model of the problem must be presented, including the optimized function and constraints. More details should be furnished.

3. What are novel aspects of the proposed approach? They must be presented in the explicit way. More details should be furnished.

4. The advantages of the proposed approach can be shown by comparison of the results with classical approaches of this field. More details should be furnished.

5. The manuscript lacks depth and fails to highlight the main focus of the considered problem. To improve the quality of the paper, it is recommended to emphasize the differences between this study and previous research and to clearly articulate the innovative contributions of this study.

6. The conclusions must be more closely directed to the novelty aspects of the paper.

6. PLOS authors have the option to publish the peer review history of their article (what does this mean?). If published, this will include your full peer review and any attached files.

Reviewer #1: No

Reviewer #2: **Yes:** Vesna Brujic-Okretic

Reviewer #3: No

Reviewer #4: No

---

## [Author Response · Author response to Decision Letter 1]

13 Apr 2026

Paper: A Streaming, Certificate-Based Reduction for Convex Hull Preservation in Planar Point Sets, PONE-D-26-02494

Dear Academic Editor and Reviewers,

We thank the Academic Editor and the reviewers for their careful reading of the manuscript and for their constructive comments. We have revised the manuscript substantially in response. The revised version includes: a clearer problem formulation, a formal hull-preservation proof, an expanded literature review, explicit novelty statements, a broader application context, order-sensitivity analysis, comparison with the Akl–Toussaint heuristic, evaluation on a large real-world dataset, strengthened runtime reporting, and improved reproducibility documentation.

Below we respond point by point to the Academic Editor and each reviewer.

Response to the Academic Editor

Comment: The authors are required to further elaborate on the problem description, algorithm design and other methodological details.

Response: We thank the Academic Editor for this guidance. In response, we revised the manuscript substantially to improve the problem description, algorithmic formulation, and methodological detail.

Changes made in manuscript:

We added a clearer Problem Formulation subsection defining the streaming setting, the retained set, the hull-preservation objective, and the associated constraints.

We clarified the conceptual relationship between exact incremental hull maintenance and the proposed relaxed streaming reduction.

We expanded the algorithmic description by introducing a precise definition of the sandwich certificate, including supporting segments and its role in safe rejection.

We added a formal Hull Preservation Guarantee section with lemmas and theorem.

We clarified the complexity model and stated the assumptions under which the O(n\log\funcapply r)bound holds.

We expanded the experimental methodology, including hardware/software environment, repetition protocol, and timing setup.

We added real-world evaluation, variability reporting, and supporting absolute runtime data.

Response to Reviewer 1

Comment: The paper needs a fully formal correctness proof, comparisons with existing convex hull filters, tests on real-world and larger datasets, an analysis of sensitivity to input order, and concrete runtime measurements to show practical benefits.

1. Formal correctness proof

Response: We agree and have added a formal proof of hull preservation. The revised manuscript now includes a dedicated proof section with explicit definitions, lemmas, and a final theorem. In particular, we formalise the sandwich certificate, prove that any discarded point lies in the convex hull of previously retained points, prove a prefix hull invariant, and then deduce exact hull preservation.

Changes made in manuscript:

Added the section Hull Preservation Guarantee.

Added a formal definition of supporting segments and the sandwich certificate.

Added Lemma 1 (certificate implies convex-hull membership).

Added Lemma 2 (prefix hull invariant).

Added Theorem 3 (exact hull preservation).

2. Comparison with existing convex hull filters

Response: We agree. The revised manuscript now includes an explicit large-scale comparison with the classical Akl–Toussaint heuristic under identical experimental conditions. This comparison is discussed both in the experimental section and in the discussion.

Changes made in manuscript:

Added Large-Scale Comparison with the Akl–Toussaint Heuristic.

Added comparative results in Figure 3, Table 1, and Table 2.

Expanded the Related Work section to position the proposed method relative to classical pre-filters.

3. Real-world and larger datasets

Response: We agree. The revised manuscript now includes experiments on a large real-world dataset using publicly available New York City Taxi & Limousine Commission Yellow Taxi Trip Data (March–May 2025). These experiments evaluate subsamples up to 500,000 points and examine the method under multiple arrival orders.

Changes made in manuscript:

Added Evaluation on Real-World GPS Data.

Added Table 2 with real-data results.

Added reproducibility information, including repository and archival DOIs.

4. Sensitivity to input order

Response: We agree. The revised manuscript now includes a dedicated order-sensitivity study across random, zigzag, and monotone orders on synthetic data, and across chronological, random, and longitude-sorted orders on the NYC dataset.

Changes made in manuscript:

Added Order Sensitivity subsection.

Added Figure 2 and corresponding discussion.

Added order-based real-data analysis in the NYC experiments section.

5. Concrete runtime measurements

Response: We agree. The revised manuscript now reports runtime speedups in the main text and includes absolute wall-clock timings in the Supporting Information. We also clarify that these measurements are based on Python reference implementations run in a consistent environment.

Changes made in manuscript:

Added runtime comparisons in Figure 3, Table 1, and Table 2.

Added Supporting Information, Table S1 with absolute runtime in seconds.

Added implementation and timing details in the experimental setup.

Response to Reviewer 2

Comment: The paper comes across as technically sound, but very narrowly focused on the particulars of the filter design to achieve the aim of the paper; it would improve the impact of this work if the context information was provided as well; e.g. how this connects with other processes involved in this type of surface reconstruction, what measurable benefits are being identified and how they can be applicable to a broader context of this topic. In addition, the statistical analysis is very sparse - it could be more informative and it should be supported by graphical representation, which is very modest at present. Also, the access to data was declared as entirely available, but it is not obvious to the reviewer how and where from the data can be accessed. As it stands, it leaves the reader wanting more information. Overall, with the appropriate amendments, I believe the paper could be publishable.

1. Broader context and application framing

Response: We agree. The revised manuscript now situates the method more clearly within broader geometric-processing workflows. We added discussion of streaming and memory-constrained pipelines, such as LiDAR acquisition, mapping, spatial sensing, surface reconstruction, and geometric summarisation. We also explain that the filter is intended as a conservative preprocessing stage rather than as a replacement for exact hull algorithms.

Changes made in manuscript:

Added the section Application Context.

Expanded the Introduction and Discussion to explain the broader setting and practical use cases.

2. Statistical analysis and graphical representation

Response: We agree. We strengthened the empirical presentation by adding a real-world dataset, additional comparison experiments, variability reporting, and supporting runtime data. The revised paper now includes three figures and two main tables, with mean ± standard deviation reporting for the NYC experiments.

Changes made in manuscript:

Added Figure 3 and Table 2.

Reported mean ± standard deviation for NYC experiments.

Added Supporting Information, Table S1 with absolute timing values.

3. Clarity of data access and availability

Response: We agree. The revised manuscript now includes a much clearer reproducibility statement. We provide the GitHub repository, as well as archived Zenodo records for both software and dataset materials. The repository includes scripts for synthetic data generation, experiment execution, figure production, and retrieval/preprocessing of the public NYC TLC datasets.

Changes made in manuscript:

Expanded the Reproducibility section.

Added permanent archival DOIs for software and dataset materials.

Clarified that the repository contains the scripts needed to reconstruct the real-data experiments.

Response to Reviewer 3

Comment: After carefully evaluating the manuscript, I did not find enough novelty or originality to support publication. The study does not demonstrate a clear conceptual, methodological, or practical advancement over existing literature. Since the manuscript lacks a strong innovative contribution and its significance appears limited, I recommend rejection.

Response: We thank the reviewer for this frank assessment. We respectfully disagree that the contribution is insufficiently novel, but we recognise that the original submission did not articulate the novelty clearly enough. In the revised manuscript, we have substantially sharpened the positioning of the work.

The novelty does not lie in proposing a new optimal convex hull algorithm. Rather, it lies in the formulation of a conservative one-pass streaming filter that:

operates strictly in streaming mode without requiring global extrema, global sorting, or multiple passes,

rejects points using a local sandwich certificate based only on incremental envelope information,

provides a formal proof of exact hull preservation, and

is empirically characterised across synthetic and real large-scale datasets, including explicit arrival-order sensitivity.

We believe this combination differentiates the present work from classical batch heuristics and from prior work on hull computation and filtering.

Changes made in manuscript:

Added an explicit Contributions list in the Introduction.

Added a clearer novelty statement near the end of the introduction.

Expanded the Related Work section, including a dedicated subsection Position of the Present Work.

Revised the Discussion and Conclusion to state more precisely what is novel and what is not claimed.

Response to Reviewer 4

Comment 1: 12 citations are clearly not enough. More references must be discussed, and please add a more detailed literature review. Author must explicitly show the place of this research in the general context of this field. More details should be furnished.

Response: We agree. The literature review has been substantially expanded. The revised manuscript now includes a broader set of classical, output-sensitive, pre-filtering, GPU, and streaming references, and a dedicated subsection positioning the present work relative to prior methods.

Changes made in manuscript:

Expanded the bibliography substantially.

Added the section Related Work, including subsections on output-sensitive methods, pre-filtering heuristics, streaming geometry, and the position of the present work.

Comment 2: Problem formulation is quite vague. The mathematical aspects of this proposal must be presented in detail to enhance the scientific rigor of the paper. The mathematical model of the problem must be presented, including the optimized function and constraints. More details should be furnished.

Response: We agree that the original problem formulation was too vague. The revised manuscript now includes a formal problem statement, explicit notation, and the constraints defining the streaming reduction task. We also clarify that the ideal objective is to minimise the retained set subject to exact hull preservation and one-pass streaming constraints, and that the proposed method is a safe conservative approximation to that constrained reduction problem.

Changes made in manuscript:

Added a detailed Problem Formulation subsection.

Defined prefixes, retained sets, hull-preservation condition, and streaming constraints.

Added the sentence describing the underlying ideal constrained-reduction objective.

Comment 3: What are novel aspects of the proposed approach? They must be presented in the explicit way. More details should be furnished.

Response: We agree. The revised manuscript now states the novel aspects explicitly in the Contributions list, in the introduction, in the related-work positioning subsection, and in the conclusion.

Changes made in manuscript:

Added a numbered Contributions list.

Added an explicit novelty statement distinguishing the method from extremal batch heuristics.

Revised the Conclusion to focus more directly on the novelty of the contribution.

Comment 4: The advantages of the proposed approach can be shown by comparison of the results with classical approaches of this field. More details should be furnished.

Response: We agree. The revised manuscript now includes direct comparison against the classical Akl–Toussaint heuristic under matched conditions, using both synthetic and real-world settings. We report reduction ratio and runtime behaviour.

Changes made in manuscript:

Added the Large-Scale Comparison with the Akl–Toussaint Heuristic section.

Added Figure 3, Table 1, and Table 2.

Comment 5: The manuscript lacks depth and fails to highlight the main focus of the considered problem. To improve the quality of the paper, it is recommended to emphasize the differences between this study and previous research and to clearly articulate the innovative contributions of this study.

Response: We agree. The revised manuscript now more clearly distinguishes the present work from prior batch heuristics and full hull-construction algorithms. The central focus is now stated more sharply: this is a conservative streaming reduction method, not a new exact hull algorithm.

Changes made in manuscript:

Revised the Introduction to sharpen the problem focus.

Added the Position of the Present Work subsection.

Revised the Discussion to emphasise the distinct operating regime and practical role of the method.

Comment 6: The conclusions must be more closely directed to the novelty aspects of the paper.

Response: We agree. The conclusion has been rewritten to emphasise the novelty, scope, and limitations of the work more directly. It now highlights the one-pass streaming nature of the method, the local sandwich certificate, the formal preservation guarantee, and the empirical order-sensitivity analysis.

Changes made in manuscript:

Revised the Conclusion to align more closely with the manuscript’s novelty claims and validated contributions.

We thank the Academic Editor and the reviewers again for their detailed and constructive comments. We believe the manuscript has been significantly strengthened as a result of these revisions, and we hope that the revised version is now suitable for publication.

Sincerely,

Oswaldo Cadenas

---

## [Decision Letter · Decision Letter 1]

3 May 2026

A Streaming, Certificate-Based Reduction for Convex Hull Preservation in Planar Point Sets

PONE-D-26-02494R1

Dear Dr. Cadenas,

We’re pleased to inform you that your manuscript has been judged scientifically suitable for publication and will be formally accepted for publication once it meets all outstanding technical requirements.

Kind regards,

Jiapeng Liu, Ph.D.

Academic Editor

PLOS One

Additional Editor Comments (optional):

Reviewers' comments:

Reviewer's Responses to Questions

**Comments to the Author**

1. If the authors have adequately addressed your comments raised in a previous round of review and you feel that this manuscript is now acceptable for publication, you may indicate that here to bypass the “Comments to the Author” section, enter your conflict of interest statement in the “Confidential to Editor” section, and submit your "Accept" recommendation.

Reviewer #1: All comments have been addressed

Reviewer #2: All comments have been addressed

2. Is the manuscript technically sound, and do the data support the conclusions?

Reviewer #1: Yes

Reviewer #2: Yes

3. Has the statistical analysis been performed appropriately and rigorously?

Reviewer #1: Yes

Reviewer #2: Yes

4. Have the authors made all data underlying the findings in their manuscript fully available?

Reviewer #1: Yes

Reviewer #2: Yes

5. Is the manuscript presented in an intelligible fashion and written in standard English?

Reviewer #1: Yes

Reviewer #2: Yes

6. Review Comments to the Author

Reviewer #1: The manuscript is well-written, clearly structured, and addresses a relevant topic. The methodology is appropriate, and the results are presented and interpreted adequately. I have no further comments. The manuscript accepted for publication.

Reviewer #2: The revised paper is of a significantly better quality answering all the points made in the comments. I believe it is ready for publication

7. PLOS authors have the option to publish the peer review history of their article (what does this mean?). If published, this will include your full peer review and any attached files.

Reviewer #1: No

Reviewer #2: **Yes:** Vesna Brujic-Okretic

---

## [Editor Report · Acceptance letter]

PONE-D-26-02494R1

PLOS One

Dear Dr. Cadenas,

I'm pleased to inform you that your manuscript has been deemed suitable for publication in PLOS One. Congratulations! Your manuscript is now being handed over to our production team.

Kind regards,

on behalf of

Dr. Jiapeng Liu

Academic Editor

PLOS One